# Controlled nonlinear magnetic damping in spin-Hall nano-devices

Boris Divinskiy[1], Sergei Urazhdin[2], Sergej O. Demokritov[1] & Vladislav E. Demidov[1]*

Large-amplitude magnetization dynamics is substantially more complex compared to the low-amplitude linear regime, due to the inevitable emergence of nonlinearities. One of the fundamental nonlinear phenomena is the nonlinear damping enhancement, which imposes strict limitations on the operation and efficiency of magnetic nanodevices. In particular, nonlinear damping prevents excitation of coherent magnetization auto-oscillations driven by the injection of spin current into spatially extended magnetic regions. Here, we propose and experimentally demonstrate that nonlinear damping can be controlled by the ellipticity of magnetization precession. By balancing different contributions to anisotropy, we minimize the ellipticity and achieve coherent magnetization oscillations driven by spatially extended spin current injection into a microscopic magnetic disk. Our results provide a route for the implementation of efficient active spintronic and magnonic devices driven by spin current.

[1] Institute for Applied Physics and Center for Nonlinear Science, University of Muenster, Corrensstrasse 2-4, 48149 Muenster, Germany. [2] Department of Physics, Emory University, Atlanta, GA 30322, USA. *email: demidov@uni-muenster.de

 1

large spin Hall effect (SHE)[1,2] exhibited by certain materials with strong spin–orbit interaction results in the generation of significant pure spin currents, enabling the implementation of a variety of efficient active magnetic nanodevices[3–5]. Pure spin currents produced by SHE enable generation of incoherent magnons[6,7] and coherent magnetic auto-oscillations[4,5,8–11], as well as excitation of propagating spin waves[12,13] in conducting and insulating magnetic materials. These applications take advantage of the compensation of the natural magnetic damping by the antidamping effect of spin current[14–16].

The antidamping torque is proportional to spin current, at small currents resulting in a linear decrease of the net effective damping. In addition to the effect on damping, spin current enhances the fluctuation amplitudes of all the spin-wave modes in the system[6]. The strongest enhancement is generally observed for the lowest-frequency mode that exhibits the smallest relaxation rate[5]. This dominant mode is expected to transition to the auto-oscillation regime at its damping compensation point. However, the increase of the amplitudes of the dynamical modes leads to their nonlinear coupling. The resulting onset of strong nonlinear relaxation of the energy and the angular momentum of the dominant mode into other spin-wave modes can be described as amplitude-dependent nonlinear damping[17–21]. Starting with the first experiments on the interaction of pure spin currents with the magnetization, this mechanism has become recognized as the main culprit generally preventing complete damping compensation and excitation of coherent magnetization auto-oscillations in spatially extended magnetic systems[6,22].

The adverse effects of nonlinear damping can be reduced either by suppressing the amplitudes of parasitic spin-wave modes, or by directly controlling the mechanisms responsible for the nonlinear mode coupling. The former approach was recently implemented by using local injection of spin current into a nanoscale region of an extended magnetic system[5,8]. In this geometry, parasitic incoherent spin waves are radiated from the localized active area, resulting in reduced nonlinear damping of the dominant mode. However, in this approach, most of the angular momentum delivered by the spin current is lost to the parasitic spin waves radiated from the active region, which requires large currents for device operation. Additionally, this approach requires that the active region is limited to nanoscale, limiting the achievable dynamical coherence and the possibilities for the magnonic device integration. Intense efforts have been dedicated to improving the dynamical coherence by utilizing arrays of mutually synchronized nanodevices, which increases the effective size of the active region[4,23], but achieving consistency and strong coupling of nanodevices have proven challenging.

Here, we demonstrate a more efficient approach based on the direct control of the mode coupling mechanisms, which allows minimization of the nonlinear damping, without constraining the geometry or the efficiency of spin current-driven nanostructures. We show experimentally and by micromagnetic simulations that the nonlinear spin-wave coupling is determined by the ellipticity of magnetization precession, which is controlled by the magnetic anisotropy. We achieve almost circular precession by tailoring the perpendicular magnetic anisotropy (PMA) of the magnetic film to compensate the dipolar anisotropy, resulting in suppression of nonlinear damping, and enabling coherent magnetization dynamics driven by the injection of spin current generated by the SHE into an extended magnetic region. The demonstrated effects are confirmed by the micromagnetic simulations, which provide additional information about the mechanisms of energy flow associated with the nonlinear damping.

## Results

**Studied system and experimental approach.** Our test devices are based on the 8 nm thick and 1.3 μm wide Pt strip, and a 5 nm thick CoNi bilayer disk with the diameter of 0.5 μm on top, Fig. 1a. The relative thicknesses of Co and Ni were adjusted so that the PMA of the disk, determined mostly by the Pt/Co and Co/Ni interface anisotropies, nearly compensates the dipolar anisotropy of the magnetic film. The saturation magnetization was $4\pi M_{CoNi} = 6.9$ kG and the PMA anisotropy field was $H_a = 6.6$ kG, as determined by separate magnetic characterization. To verify that the demonstrated effects originate from PMA, we have also studied a control sample utilizing a 5 nm thick Permalloy (Py) disk with negligible PMA, instead of the CoNi.

Because of SHE in Pt, current $I$ produces an out-of-plane spin current $I_S$, which is injected into the ferromagnet (inset in Fig. 1a), exerting antidamping spin torque on its magnetization $\mathbf{M}$[14]. To maximize the antidamping effect of spin current, the magnetization in the studied devices was saturated by the in-plane static magnetic field $\mathbf{H_0} = 1$–2 kOe applied perpendicular to the direction of the current flow.

At sufficiently large $I$, spin current-induced torque may be expected to completely compensate the natural damping, resulting in the excitation of coherent magnetization auto-oscillations. However, experiments have shown that this does not occur if spin current is injected into an extended magnetic region. Instead, the effects of spin current saturate due to the onset of nonlinear damping, and compensation is never achieved[6].

The effects of magnetic anisotropy on the magnetization precession characteristics are illustrated in Fig. 1b, c. The precessing magnetization vector $\mathbf{M}$ induces dynamic demagnetizing field $\mathbf{h_d}$ antiparallel to the out-of-plane component $\mathbf{M_x}$ of magnetization, resulting in an elliptical precession trajectory with the short axis normal to the film plane (Fig. 1b). The elliptical precession is accompanied by the oscillation of the component of magnetization parallel to the precession axis $m_z^{2\omega}$, at twice the frequency of precession. This oscillation acts as a parametric pump[24,25] that drives energy transfer from the dominant excited spin-wave mode into other modes, resulting in nonlinear damping of the former.

In contrast to $\mathbf{h_d}$, the effective dynamical field $\mathbf{h_{PMA}}$ associated with PMA is parallel to $\mathbf{M_x}$. If the magnitude of $\mathbf{h_{PMA}}$ is close to that of $\mathbf{h_d}$, the two fields compensate each other, and precession becomes circular (Fig. 1c). As follows from the arguments given above, nonlinear damping is expected to become suppressed in films with PMA compensating dipolar anisotropy.

We experimentally characterize the effects of PMA on the current-induced magnetization dynamics by using micro-focus Brillouin light scattering (BLS) spectroscopy[26]. The probing laser light is focused on the surface of the magnetic disk (Fig. 1a), and the modulation of the scattered light by high-frequency magnetization oscillations is detected. The BLS signal intensity is proportional to the intensity of magnetization oscillations at the selected frequency.

**Magnetooptical measurements.** Fig. 1d, e show the BLS spectra of magnetic oscillations detected in Py (Fig. 1d) and CoNi (Fig. 1e) disks with current $I$ close to the critical value $I_C$, at which the spin current is expected to completely compensate the natural linear magnetic damping (see Supplementary Note 1 and Supplementary Figure 1 for the determination of the critical currents).

At $I < I_C$, both Py and CoNi exhibit similar increases of the BLS intensity with current, as expected due to the enhancement of magnetic fluctuations by the spin-transfer torque[27]. However, the

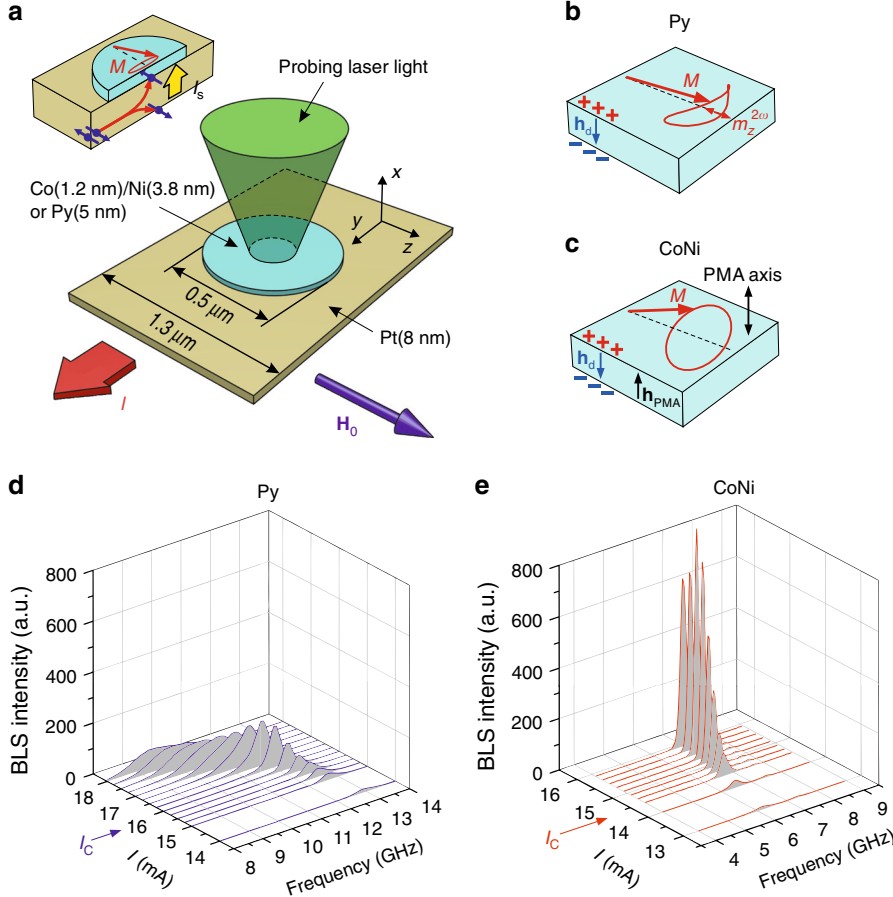

**Fig. 1** Schematics of the experiment and spectra of magnetic oscillations. **a** Layout of the test devices. Magnetic disks are fabricated on top of a Pt strip either from Py, or from the Co/Ni bilayer with PMA tailored to compensate the dipolar anisotropy of the film. Inset illustrates the device operation principle, based on the injection into the ferromagnetic disk of pure spin current generated due to the SHE in Pt. **b** The ellipticity of the magnetization precession in Py is caused by the dipolar anisotropy. **c** In CoNi, the ellipticity is minimized due to PMA compensating the dipolar anisotropy. **d**, **e** BLS spectra of magnetic oscillations vs current for Py and CoNi disks, respectively. $I_C$ marks the critical current, at which the spin current is expected to completely compensate the natural linear magnetic damping. The data were recorded at $H_0 = 2.0$ kOe

behaviors diverge at $I > I_C$. In the Py disk, the intensity of fluctuations saturates, while their spectral width significantly increases (Fig. 1d). In contrast, a narrow intense peak emerges in CoNi, marking a transition to the auto-oscillation regime (Fig. 1e). These results indicate that the phenomena preventing the onset of auto-oscillations in the Py disk are suppressed in CoNi.

The differences between the two systems are highlighted by the quantitative analysis of their characteristics, Fig. 2. At $I < I_C$, the BLS intensity increases (Fig. 2a), while the spectral width of fluctuations decreases due to the reduced effective damping (Fig. 2b), following the same dependence for both Py and CoNi. In Py, the peak intensity starts to decrease at $I > I_C$, while the spectral width rapidly increases, indicating the onset of nonlinear damping. In contrast, in CoNi the intensity rapidly increases at $I > I_C$, while the spectral linewidth continues to decrease. We note that the BLS spectra are broadened by the finite frequency resolution of the technique, increasing the measured values particularly for small linewidths. At large currents, the BLS intensity in CoNi somewhat decreases and the spectral width increases, indicating an onset of higher-order nonlinear processes that cannot be completely avoided in real systems.

The two systems also exhibit a qualitatively different dependence of the BLS peak frequency on current—the nonlinear frequency shift (Fig. 2c). For CoNi, the frequency slightly increases with current, while for Py, it exhibits a redshift that

becomes increasingly significant above $I_C$. The large frequency nonlinearity in Py is likely associated with the nonlinear excitation of a broad spectrum of spin-wave modes, which is directly related to the nonlinear suppression of auto-oscillation, as discussed in detail below.

The effects of the nonlinear damping can be also clearly seen in the time-domain BLS measurements. In these measurements, the current was applied in pulses with the duration of 1 μs and period of 5 μs, and the temporal evolution of the BLS intensity was analyzed. Figure 3 shows the temporal traces of the intensity recorded for the Py and CoNi disks at $I = 1.07I_C$, corresponding to the maximum intensity achieved in CoNi sample (Fig. 2a). For the CoNi disk, the intensity monotonically increases and then saturates. In contrast, for the Py disk, the intensity saturates at a much lower level shortly after the start of the pulse, followed by a gradual decrease over the rest of the pulse duration, indicating the onset of energy flow into other spin-wave modes.

**Micromagnetic simulations**. The mechanisms underlying the observed behaviors are elucidated by the micromagnetic simulations, which were performed using the MuMax3 software[28]. The linear spin-wave dispersions, calculated using the small-amplitude limit $M_y^{max}/M = 0.01$, are qualitatively similar for Py and CoNi (symbols in Fig. 4a, b). The two branches

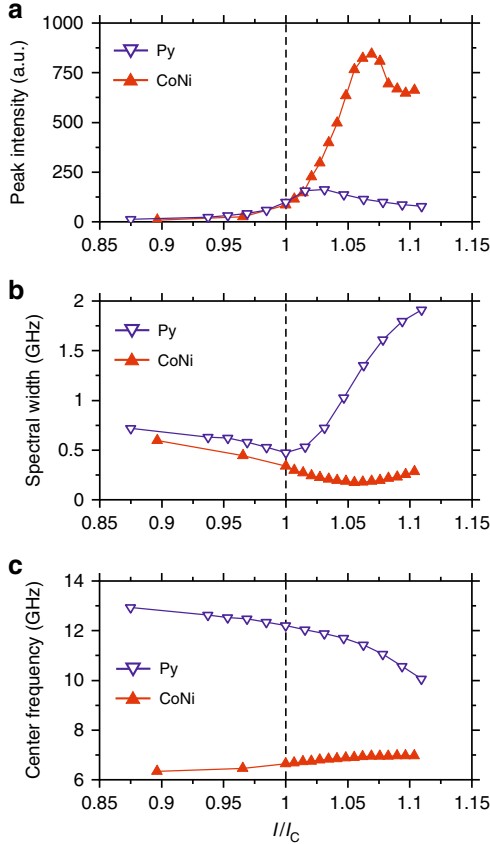

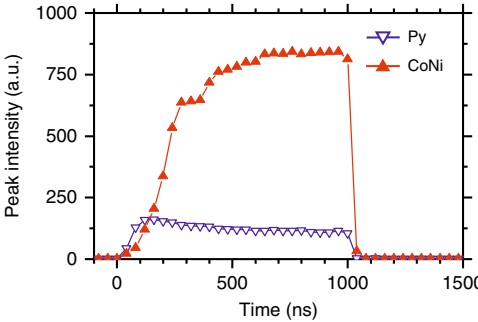

**Fig. 2** Characterization of the current-induced magnetization dynamics. **a** Maximum intensities of the BLS spectra vs current. **b** Current dependences of the spectral width of the BLS peaks at half the maximum intensity. **c** Center frequency of the detected spectral peaks vs current. Symbols are the experimental data, lines are guides for the eye. The data were recorded at $\mathbf{H_0} = 2.0$ kOe

**Fig. 3** Evolution of the nonlinear damping in the time domain. Time dependence of the peak BLS intensity in response to the 1 µs long pulse of current obtained for the Py and CoNi disks, as labeled. The data were recorded at $\mathbf{H_0} = 2.0$ kOe and $I = 1.07 I_C$

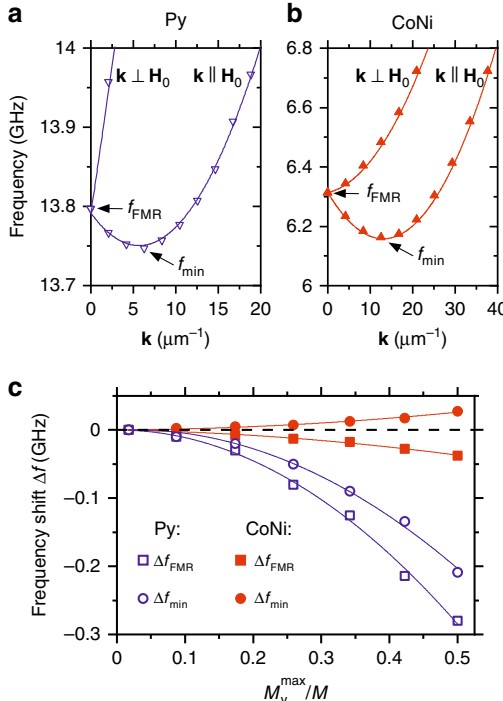

**Fig. 4** Calculated spin-wave dispersion spectra. **a**, **b** Dispersion spectra for the Py and CoNi films, respectively, calculated in the small-amplitude linear regime. $f_{FMR}$ and $f_{min}$ label the frequencies of the quasi-uniform FMR and of the lowest-frequency spin-wave mode, respectively. Symbols are the results of micromagnetic simulations, curves—calculations based on the analytical theory. **c** Dependences of the characteristic frequencies $f_{FMR}$ and $f_{min}$ on the normalized precession amplitude. Symbols are the results of micromagnetic simulations, curves—guides for the eye. All calculations were performed at $\mathbf{H_0} = 2.0$ kOe

corresponding to spin waves propagating perpendicular and parallel to the static field $\mathbf{H_0}$ merge at the wavevector $\mathbf{k} = 0$, at the frequency $f_{FMR}$ of the uniform-precession ferromagnetic resonance (FMR). The frequency of the branch with $\mathbf{k} \perp \mathbf{H_0}$ monotonically increases with $\mathbf{k}$, while the branch with $\mathbf{k} \| \mathbf{H_0}$ exhibits a minimum $f_{min}$ at finite $\mathbf{k}$ due to the competition between the dipolar and the exchange interactions. The frequencies obtained

from the simulations are in good agreement with the results of calculations using analytical spin-wave theory (solid curves in Fig. 4a, b)[29].

The qualitative differences between the nonlinear characteristics of the two systems are revealed by the dependence of frequency on the amplitude of magnetization oscillations, Fig. 4c. In the Py film, both $f_{FMR}$ and $f_{min}$ exhibit a strong negative nonlinear frequency shift. In contrast, in the CoNi film the frequency $f_{FMR}$ slightly decreases, while the frequency $f_{min}$ increases with increasing amplitude. This difference allows us to identify the auto-oscillation mode excited in the CoNi sample. Since the experimentally observed oscillation frequency for CoNi sample increases with increasing amplitude (Fig. 2c), we conclude that current-induced auto-oscillations correspond not to the quasi-uniform FMR mode, but rather to the lowest-frequency spin-wave mode. This conclusion is in agreement with the recent experimental observations[30], which showed that the injection of pure spin current results in the accumulation of magnons in the state with the lowest frequency.

We now analyze the relationship between the dispersion characteristics and the nonlinear damping effects. The lowest-frequency state at $f_{min}$ is non-degenerate in both Py and CoNi (Fig. 4a, b). The absence of degeneracy is commonly viewed as a sufficient condition for the suppression of nonlinear damping, since it prohibits resonant four-wave interactions[10]. However, this view is inconsistent with our experimental results, Fig. 1d, e, as also confirmed by the micromagnetic simulations illustrated in Fig. 5. In these simulations, we use artificially small linear

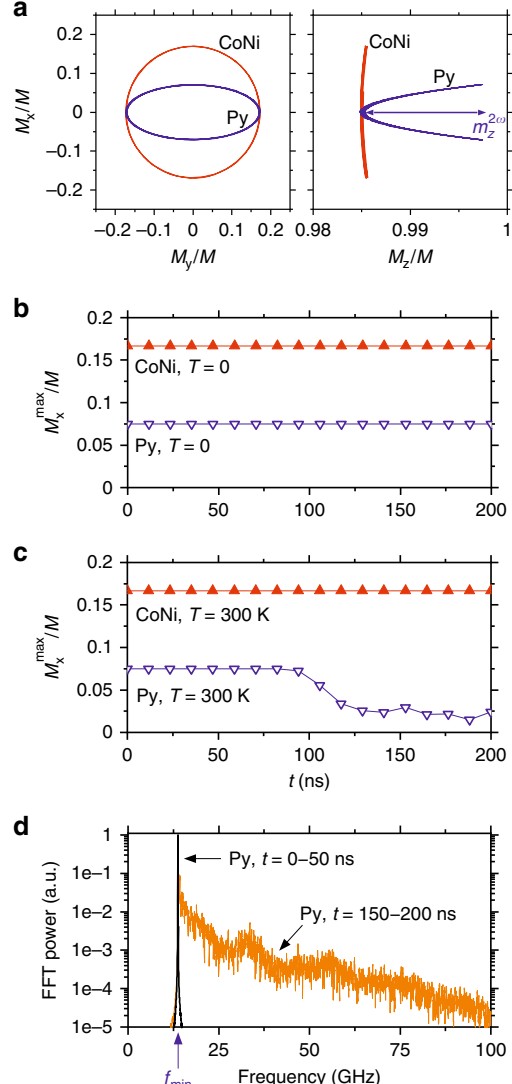

**Fig. 5** Analysis of nonlinear damping based on the micromagnetic simulations. **a** Calculated magnetization trajectories for the lowest-frequency spin-wave states in Py and CoNi, as labeled. $m_z^{2\omega}$ labels the double-frequency dynamic component of magnetization, which serves as a parametric pumping source for the nonlinear spin-wave excitation. **b, c** Temporal evolution of the free precession amplitude starting with a large initial amplitude at $t = 0$, at $T = 0$ (**b**) and $T = 300$ K (**c**). The simulations were performed with negligible linear damping, emulating the damping compensation by the spin current. **d** Fourier spectra of magnetization oscillations in Py before ($t = 0$-50 ns) and after ($t = 150$-200 ns) the onset of nonlinear damping

damping to emulate damping compensation by the spin current, and analyze the dynamics of the lowest-frequency mode excited at time $t = 0$. Figure 5a shows the projections of the magnetization precession trajectories on the $M_y$–$M_x$ and $M_z$–$M_x$ planes, for a relatively large precession amplitude $M_y^{max}/M = 0.17$. As expected, the precession is nearly circular in CoNi, and elliptical in Py. The ellipticity in Py results in the oscillation of the projection $m_z^{2\omega}$ of magnetization on the equilibrium direction at twice the oscillation frequency, which plays the role of a parametric pump for other spin-wave modes. In simulations performed for zero temperature (Fig. 5b), precession initiated at $t = 0$ continues indefinitely, i.e. energy is not transferred to other

modes. This result is consistent with the parametric mechanism of mode coupling, which requires non-zero amplitudes of all the involved modes. In contrast to $T = 0$, at finite temperatures all the spin-wave modes have non-zero amplitudes due to thermal fluctuations, enabling their parametric excitation. In the simulations performed at $T = 300$ K, the amplitude of precession excited in Py at $t = 0$ abruptly drops at about 100 ns and continues to decrease thereafter, indicating the onset of nonlinear damping (Fig. 5c). Spectral analysis confirms that the initially monochromatic oscillation at frequency $f_{min}$ transitions to a broad spectrum of spin-wave modes excited at longer times due to their nonlinear coupling to the initially excited mode (Fig. 5d). We emphasize that this nonlinear coupling must be non-resonant[18,31], since it cannot be described in terms of energy- and momentum-conserving magnon-magnon interactions[25].

The parametric mechanism of the nonlinear mode coupling is confirmed by the simulation results for the CoNi film, where the oscillations of the longitudinal magnetization component are negligibly small, and the precession amplitude remains constant (Fig. 5c). These results clearly show that the compensation of the precession ellipticity by the PMA enables suppression of the nonlinear damping, supporting our interpretation of the experimental findings.

## Discussion

The simulations described above were performed for a model system—an extended magnetic film characterized by a continuous spin-wave spectrum. The generality of the nonlinear damping mechanism revealed by these simulations, and its relevance to our experimental results, was confirmed by additional simulations performed for 0.5 μm Py and CoNi disks matching those studied in our experiments. The temporal evolution of the lowest-frequency modes is very similar to that obtained for extended films (see Supplementary Figure 2). However, the spectrum excited due to the nonlinear damping of the lowest-frequency mode becomes discrete, consistent with the quantization of spin-wave spectrum expected for a confined magnetic system.

Finally, we discuss the consequences of residual precession ellipticity, which can result from incomplete compensation of the dipolar anisotropy by PMA. To characterize these effects, we performed additional measurements for CoNi samples with the anisotropy field deviating from the saturation magnetization by about 10% in both directions (Supplementary Figure 3). For these samples, the transition to the auto-oscillation regime becomes noticeably suppressed by the nonlinear damping, confirming the importance of precise suppression of the precession ellipticity to achieve auto-oscillations in an extended magnetic region (Supplementary Note 2).

In conclusion, our experiments and simulations show that the adverse nonlinear damping can be efficiently suppressed by minimizing the ellipticity of magnetization precession, using magnetic materials where in-plane dipolar anisotropy is compensated by the PMA. This allows one to achieve complete compensation of the magnetic damping, and excitation of coherent magnetization auto-oscillations by the spin current without confining the spin current injection area to nanoscale. Our findings open a route for the implementation of spin-Hall oscillators capable of generating microwave signals with technologically relevant power levels and coherence, circumventing the challenges of phase locking a large number of oscillators with nanoscale dimensions. They also provide a route for the implementation of spatially extended amplification of coherent propagating spin waves, which is vital for the emerging field of

magnonics utilizing these waves as the information carrier. The proposed approach can also facilitate the experimental realization of spin current-driven Bose–Einstein condensation of magnons, which has not be achieved so far due to the detrimental effects of nonlinear damping[30].

## Methods

**Sample fabrication and characterization.** The samples were fabricated on annealed sapphire substrates with pre-patterned Au electrodes. First, 1.3-μm-wide Ta(4)Pt(8)FM(5)Ta(3) strips connected to the Au electrodes were fabricated by a combination of e-beam lithography and high-vacuum sputtering. Here, thicknesses are in nanometers, and the ferromagnetic layer FM(5) is either Co(1.2)/Ni(3.8) or Py(5). The bottom Ta(4) was used as a buffer layer, and the top Ta(3) as a capping layer to prevent oxidation of the magnetic structures. A circular 500 nm $AlO_x(10)$ mask was then defined by Al evaporation in 0.02 mTorr of oxygen, followed by Ar ion milling timed to remove the part of the multilayer down to the top of the Pt(8) layer. Finally, the structures were passivated with an $AlO_x(10)$ capping layer.

The values of the saturation magnetization $4\pi M_{CoNi} = 6.9$ kG and $4\pi M_{Py} = 10.1$ kG were determined from vibrating-sample magnetometry measurements. The PMA anisotropy field $H_a = 6.6$ kG of CoNi was obtained from the FMR frequency and the known magnitude of the magnetization.

**Measurements.** All measurements were performed at room temperature. In micro-focus BLS measurements, the probing laser light with the wavelength of 532 nm and the power of 0.1 mW was focused into a diffraction-limited spot on the surface of the magnetic disk by using a high-numerical-aperture 100x microscope objective lens. The position of the probing spot was actively stabilized by using custom-designed software providing long-term spatial stability of better than 50 nm. The spectrum of light inelastically scattered from magnetization oscillations was analyzed by a six-pass Fabry–Perot interferometer TFP-2HC (JRS Scientific Instruments, Switzerland). The obtained BLS intensity at a given frequency is proportional to the square of the amplitude of the dynamic magnetization at this frequency. Therefore, BLS spectra directly represent the spectral distribution of the magnetization oscillations in the magnetic disk.

**Micromagnetic simulations.** The micromagnetic simulations were performed by using the software package MuMax3[28]. We modeled the magnetization dynamics in a 10 μm × 10 μm large and 5 nm thick magnetic film. The computational area was discretised into 5 nm ×5 nm ×5 nm cells. Periodic boundary conditions at all lateral boundaries were used. Magnetization dynamics was excited by initially deflecting magnetic moments from their equilibrium orientation in the plane of the film. The deviation was either spatially uniform or periodic. By analyzing the free dynamics of magnetization, the frequency corresponding to the given spatial period was determined, which allowed us to reconstruct the frequency vs wavevector dispersion curves (see Supplementary Figure 4). By varying the angle of the initial deflection, we determined the dependence of the frequency on the oscillation amplitude and analyzed the instability of precession caused by the nonlinear damping at large amplitudes (see Supplementary Figure 5). Simulations were performed using artificially small Gilbert damping parameter $10^{-12}$, which emulates the conditions of the compensation of the linear damping by the pure spin current. The values of the saturation magnetization $4\pi M_{CoNi} = 6.9$ kG and $4\pi M_{Py} = 10.1$ kG were determined from the vibrating-sample magnetometry measurements. The PMA anisotropy constant of CoNi film 0.181 J m$^{-3}$ was calculated based on the experimentally determined values of the anisotropy field and the magnetization. Standard values of the exchange stiffness constant of 13 pJ m$^{-1}$ and 9 pJ m$^{-1}$ were used for Py and CoNi, respectively.

## Code availability

The open-source software package MuMax3 used in this study is released under the GPLv3 license and is available at https://mumax.github.io/.

## Data availability

The data that support the findings of this study are available from the corresponding author upon reasonable request.

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

## Acknowledgements

The authors acknowledge support from Deutsche Forschungsgemeinschaft (Project No. 423113162) and the National Science Foundation of USA.

## Author contributions

B.D. and V.E.D. performed measurements and data analysis. B.D. additionally performed micromagnetic simulations. S.U. designed and fabricated the samples. S.U., S.O.D., and V.E.D. managed the project. All authors co-wrote the manuscript.

## Competing interests

The authors declare no competing interests.
