## [Peer Review File · Nature Communications]

Reviewers' comments:

Reviewer #1 (Remarks to the Author):

This manuscript by B. Divinskiy et al. reports a novel route for the suppression of nonlinear damping by controlling the ellipticity of magnetization precession, which allows achieving coherent magnetization auto-oscillations induced by the spin Hall effect without confining the active area. Through tuning the relative strength of dipolar anisotropy and perpendicular magnetic anisotropy (PMA) in CoNi to reach compensation, the authors observed a reduced spectral width with an enhancement of peak intensity in BLS spectra, as the value of injected current exceeds the critical one, corresponding to the excitation of auto-oscillations. The micromagnetic simulations further supported this idea, whose underlying mechanism is related to the lowest frequency spin wave mode. This finding may facilitate the integration of magnonic devices for applications, and the manuscript is clearly organized and interesting. However, I still have some questions that need to be sufficiently addressed before I recommend the publication in Nature Communications.

Comments:

1. Since the effects of spin torque on magnetization dynamics are non-selective, the injection of spin current into magnetic film would simultaneously enhance all dynamical modes, which results in nonlinear scattering processes, preventing the transition to auto-oscillation regime. One possible strategy to avoid the detrimental effects of nonlinear damping is to implement mode-selective radiation losses by localizing the spin current injection, as proposed for the geometry of planar nano-gap oscillator. In the small gap region, the enhanced high frequency spin wave quickly escapes from the active region, while the radiation loss of the enhanced low frequency spin wave is minimal. This manuscript demonstrates that the circular magnetization precession can suppress the nonlinear damping without confining the active area, having a similar consequence on realizing auto-oscillation to the nano-gap oscillator, but the full interpretation is lacking. Besides, the values of critical current for both Py and CoNi are quite close as given in the supplementary material. Therefore, the authors should discuss the possible mechanism accounting for the circular magnetization precession promoted auto-oscillation more in details, regarding to the enhancement of spin wave modes via the injection of spin current.

2. If the auto-oscillation can be realized by controlling the ellipticity of magnetization precession, an intermediate state should be obtained through making the PMA somewhat larger or smaller than the dipolar anisotropy, and further verify this consideration. Thus, I ask the authors to check such intermediate state of elliptical magnetization precession by performing the BLS measurements on an additional sample. This state might be achieved by adjusting the relative thickness of Co and Ni in CoNi film.

3. In Figs. 4b and 4c, the temporal evolution of the free precession amplitude was given, showing distinct characteristics associated with the magnetization relaxation at 300 K, while the experimental evidences are absent. Since the time resolved BLS measurements on the temporal evolution of effective magnetization and BLS integral intensity are an efficient way to characterize the nonlinear coupling among different modes, it is necessary to perform such measurements with the value of applied current larger than the critical one while smaller than the onset of BLS intensity decrease in CoNi, which may provide more evidences from experiments.

4. In this manuscript, the mechanism related to the lowest frequency spin wave mode was described based on the micromagnetic simulations. Thus, some details on the simulations should be given probably in the supplementary material, such as the curves of theoretical calculations in Figs. 3a and 3b, and the temporal evolution of the free precession amplitude in Figs. 4(b-d).

Reviewer #2 (Remarks to the Author):

The authors claim a solution for a big challenge which concerns excitation of coherent magnetization auto-oscillations driven by the injection of spin current into spatially extended magnetic regions. The authors suggest control of nonlinear damping by avoiding magnetization precession with pronounced ellipticity. The experimental data are original and very interesting. They provide a novel route for the implementation of active spintronic and magnonic devices. However the authors present experimental data taken at only one single magnetic field value on disks with a diameter of 500 nm. Micromagnetic simulations were performed on a sample with significantly different lateral dimensions which were much larger and were 10000 nm x 10000 nm. Furthermode periodic boundary conditions were applied. Injected currents were not considered in the simulations. Instead magnetization vectors were misaligned in specific arrangements. The simulated frequencies do not agree with the frequency regime observed in experiments. Discrete modes are seen in the spectra of the CoFeB disk which are not contained in the simulations. At this stage conclusions drawn from simulations do not consolidate new findings concerning spin injection. Ellipticity is already known as dominating factor for coupling to spin wave modes (see textbook in Ref. [24] of the present manuscript).

The authors do not consider that anisotropy in PMA materials often varies locally which induces additional magnon scattering. Sample boundaries modify the spin wave modes which contribute efficiently to nonlinear damping, scattering at boundaries increases threshold fields for nonlinear effects. These aspects were not discussed when comparing data and simulations. In Fig. 3 b the orientation of the magnetization vector in the film is not clear. The orientation is decisive for

magnetization dynamics and should be different for the simulated large magnet with periodic boundary conditions compared with the real CoFeB disk with a diameter of only 500 nm. In the disk there is an inhomogeneous demagnetization field which is not the case for the simulations. To raise broad interest simulations should be conducted which model the experiment realistically. Experimental results taken for more than one specific field H are important as nonlinear damping depends critically on H .

Reviewer #3 (Remarks to the Author):

This is a very nice manuscript in which a solution to excite magnetization auto-oscillation from Spin Hall effect by suppressing non-linear damping is provided. The author have provided clear experimental evidence of complete damping compensation from spin current, and verified with micromagnetics simulation. The investigation clearly identify the cause to non-linear damping and the results will be very useful to develop functional spin nano-oscillators. The experiment and simulation are well designed and carried out. The manuscript is clear and is free of mistakes. I recommend the publication of this manuscript.

Response to Reviewer #1

The Reviewer writes:

This manuscript by B. Divinskiy et al. reports a novel route for the suppression of non-linear damping by controlling the ellipticity of magnetization precession, which allows achieving coherent magnetization auto-oscillations induced by the spin Hall effect without confining the active area. Through tuning the relative strength of dipolar anisotropy and perpendicular magnetic anisotropy (PMA) in CoNi to reach compensation, the authors observed a reduced spectral width with an enhancement of peak intensity in BLS spectra, as the value of injected current exceeds the critical one, corresponding to the excitation of auto-oscillations. The micromagnetic simulations further supported this idea, whose underlying mechanism is related to the lowest frequency spin wave mode. This finding may facilitate the integration of magnonic devices for applications, and the manuscript is clearly organized and interesting. However, I still have some questions that need to be sufficiently addressed before I recommend the publication in Nature Communications.

Reply:

We thank the Reviewer for the positive evaluation of our work. Below, we respond in detail to all the Reviewer's questions, and describe how they have been addressed in the revised manuscript. We hope that the Reviewer will find our answers and the changes made in the manuscript satisfactory, and will recommend publication of the revised manuscript.

The Reviewer writes:

1. Since the effects of spin torque on magnetization dynamics are non-selective, the injection of spin current into magnetic film would simultaneously enhance all dynamical modes, which results in nonlinear scattering processes, preventing the transition to auto-oscillation regime. One possible strategy to avoid the detrimental effects of nonlinear damping is to implement mode-selective radiation losses by localizing the spin current injection, as proposed for the geometry of planar nano-gap oscillator. In the small gap region, the enhanced high frequency spin wave quickly escapes from the active region, while the radiation loss of the enhanced low frequency spin wave is minimal. This manuscript demonstrates that the circular magnetization precession can suppress the nonlinear damping without confining the active area, having a similar consequence on realizing auto-oscillation to the nano-gap oscillator, but the full interpretation is lacking. Besides, the values of critical current for both Py and CoNi are quite close as given in the supplementary material. Therefore, the authors should discuss the possible mechanism accounting for the circular magnetization precession promoted auto-oscillation more in details, regarding to the enhancement of spin wave modes via the injection of spin current.

Reply:

We thank the Reviewer for pointing out this issue. We agree that the original manuscript did not include a sufficiently thorough discussion of the role of simultaneous enhancement of different modes by the spin current. In the revised manuscript, we have rewritten the introduction (pages 2-3) to make this role clearer. We now describe the effects of spin current on the mode amplitudes, as summarized by the Reviewer. We also clearly state that the nonlinear damping (nonlinear scattering) can be reduced either by suppressing the enhancement of the amplitudes of parasitic spin-wave modes, or by directly controlling the mechanisms responsible for the nonlinear mode coupling. The previously demonstrated systems utilized the former approach. While this approach was found to be efficient, it requires a combination of large size of the magnetic system to provide radiative damping, and a nanoscale active area. These

requirements are not well-suited for device optimization. In contrast, in the present study we demonstrate a route to directly control the mechanisms responsible for mode coupling. Our results show that the main coupling mechanism is the parametric coupling, which requires non-zero ellipticity of magnetization precession. Correspondingly, by suppressing the ellipticity, one can suppress the interactions among the modes, even if their amplitudes are large. In a system with circular precession, the spin current still enhances all the modes. However, the dominant mode possessing the smallest linear relaxation rate, which is enhanced most efficiently, does not exchange energy and angular momentum with the parasitic modes.

The Reviewer writes:

2. If the auto-oscillation can be realized by controlling the ellipticity of magnetization precession, an intermediate state should be obtained through making the PMA somewhat larger or smaller than the dipolar anisotropy, and further verify this consideration. Thus, I ask the authors to check such intermediate state of elliptical magnetization precession by performing the BLS measurements on an additional sample. This state might be achieved by adjusting the relative thickness of Co and Ni in CoNi film.

Reply:

We agree with the Reviewer that the information about the intermediate states of elliptical precession can be interesting to the reader. In fact, when optimizing our CoNi samples to achieve compensation, we also performed measurements for samples with incomplete compensation. Complying with the Reviewer's comment, we have added Supplementary Fig. 3 showing the evolution of the peak intensity with current for three different CoNi samples with PMA anisotropy fields of 6.3, 6.6, and 7.6 kOe. In the first sample, the PMA anisotropy is about 9% smaller than the dipolar anisotropy of 6.9 Oe, while in the third sample it is about 10% larger. The second sample with almost complete compensation is discussed in the main text. The data in the Supplementary Fig. 3 clearly show that the transition to the auto-oscillation regime becomes suppressed by the nonlinear damping in both samples where the PMA anisotropy is sufficiently different from the saturation magnetization. We have added a short discussion to this effect on page 10 of the revised manuscript.

The Reviewer writes:

3. In Figs. 4b and 4c, the temporal evolution of the free precession amplitude was given, showing distinct characteristics associated with the magnetization relaxation at 300 K, while the experimental evidences are absent. Since the time resolved BLS measurements on the temporal evolution of effective magnetization and BLS integral intensity are an efficient way to characterize the nonlinear coupling among different modes, it is necessary to perform such measurements with the value of applied current larger than the critical one while smaller than the onset of BLS intensity decrease in CoNi, which may provide more evidences from experiments.

Reply:

We thank the Reviewer for this suggestion. We agree that time-resolved measurements provide direct experimental evidence for the presence/absence of the nonlinear damping. Following the Reviewer's suggestion, we have performed additional time-resolved BLS measurements and added the results in the new Fig. 3 in the revised manuscript (see also the added discussion on page 7). As seen from Fig. 3, in the CoNi disk, the intensity monotonically increases with time, and then saturates. In contrast, as expected for the effects of the nonlinear damping, in the Py disk, the growth of the intensity stops at much smaller levels already at the beginning of the pulse and is then followed by a gradual decrease over the rest of the pulse duration, indicating the onset of the energy flow into other spin-wave modes. We note that the

observed decrease is not as abrupt as in simulations, because the energy flow into other modes is counterbalanced by the continuous spin current injection.

The Reviewer writes:

4. In this manuscript, the mechanism related to the lowest frequency spin wave mode was described based on the micromagnetic simulations. Thus, some details on the simulations should be given probably in the supplementary material, such as the curves of theoretical calculations in Figs. 3a and 3b, and the temporal evolution of the free precession amplitude in Figs. 4(b-d).

Reply:

Complying with the Reviewer's comment, we have added in the Supplementary Information an additional description of the simulations, illustrated with Supplementary Figs. 4 and 5. In particular, we show an example of the determination of the frequency corresponding to the spin wave with a particular wavevector (data shown in Figs. 3a and 3b), and an example of the analysis of the time evolution of the free precession amplitude (data shown in Figs. 4b-4d).

Response to Reviewer #2

The Reviewer writes:

The authors claim a solution for a big challenge which concerns excitation of coherent magnetization auto-oscillations driven by the injection of spin current into spatially extended magnetic regions. The authors suggest control of nonlinear damping by avoiding magnetization precession with pronounced ellipticity. The experimental data are original and very interesting. They provide a novel route for the implementation of active spintronic and magnonic devices.

Reply:

We thank the Reviewer for the positive evaluation of our work and for the appreciation of the novelty and the importance of our findings. As far as we can see, the Reviewer does not doubt the validity of the experimental findings and their interpretation. His/her main concerns are about the details of simulations, which were included in the original manuscript to give the reader a general idea about the mechanisms of the nonlinear damping and the route to control it by suppressing the ellipticity. Below, we respond in detail to all the Reviewer's concerns, and describe how they have been addressed in the revised manuscript. We hope that the Reviewer will find our explanations satisfactory and will recommend publication of the revised manuscript.

The Reviewer writes:

However the authors present experimental data taken at only one single magnetic field value on disks with a diameter of 500 nm. Micromagnetic simulations were performed on a sample with significantly different lateral dimensions which were much larger and were 10000 nm x 10000 nm. Furthermode periodic boundary conditions were applied. Injected currents were not considered in the simulations. Instead magnetization vectors were misaligned in specific arrangements. The simulated frequencies do not agree with the frequency regime observed in experiments. Discrete modes are seen in the spectra of the CoFeB disk which are not contained in the simulations. At this stage conclusions drawn from simulations do not consolidate new findings concerning spin injection. Ellip-

tivity is already known as dominating factor for coupling to spin wave modes (see textbook in Ref. [24] of the present manuscript).

Reply:

Before we address these concerns in detail, we would like to note that the simulations presented in the original manuscript were not intended to reproduce the behaviors of the experimental system in all their complexity, but rather to elucidate the fundamental mechanisms responsible for the non-resonant nonlinear damping. In principle, this goal could be also achieved without simulations, if one could controllably excite in the experiment specific dynamic modes with predefined amplitudes, and measure the nonlinear redistribution of energy with high temporal and spectral resolution. Since this is not possible in practice, we performed simulations using a simplified system that captured the essential aspects of the mechanisms underlying the nonlinear damping. For example, the effects of the spin injection were taken into account not by directly using the spin-transfer torque term, but by setting the conditions corresponding to the known effects of spin current - compensated damping and finite oscillation amplitudes of all the dynamic modes. As a result, we were able to gain insight into the mechanism of the nonlinear damping of the lowest-frequency mode known to be most efficiently excited by the spin current, and determine the approaches to controlling this mechanism. We emphasize, that until our work, this mechanism has remained unclear. In particular, the usual resonant nonlinear scattering processes like three- and four-magnon interactions, discussed in the textbook Ref. [24] mentioned by the Reviewer, are prohibited for the lowest-frequency mode by the conservation laws.

To comply with the Reviewer's comment and to demonstrate that the observed fundamental mechanisms are at play in the confined samples, we performed additional simulations taking into account the finite size of the disks. We added the corresponding data to the Supplementary Information (Supplementary Fig. 2). We have also added a discussion on the influence of the lateral confinement on page 9 of the revised manuscript.

The Reviewer writes:

The authors do not consider that anisotropy in PMA materials often varies locally which induces additional magnon scattering. Sample boundaries modify the spin wave modes which contribute efficiently to nonlinear damping, scattering at boundaries increases threshold fields for nonlinear effects. These aspects were not discussed when comparing data and simulations.

Reply:

We agree with the Reviewer that imperfections and boundary effects are important for the full quantitative understanding of the dynamical behaviors. However, the main purpose of our simulations is to elucidate the general origin of the nonlinear damping and the possibility to control it by minimizing the ellipticity. Furthermore, we do not expect our conclusions to be invalidated by the boundary effects and modest local variations of anisotropy, because of the strong limitations on resonant magnon scattering imposed by the conservation laws. For instance, simulations for a microscopic disk presented in the revised manuscript demonstrate the same effects of ellipticity as continuous films. A more detailed analysis of the effects of imperfections on nonlinear magnon scattering in spin current-driven structures is clearly beyond the scope of the present work. We hope that these effects will be thoroughly addressed in the future theoretical and experimental studies spurred by our results.

The Reviewer writes:

In Fig. 3 b the orientation of the magnetization vector in the film is not clear. The orientation is decisive for magnetization dynamics and should be different for the simulated large magnet with periodic boundary conditions compared with the real CoFeB disk with a diameter of only 500 nm. In the disk there is an inhomogeneous demagnetization field which is not the case for the simulations. To raise broad interest simulations should be conducted which model the experiment realistically.

Reply:

We have performed additional simulations for magnetic structures with dimensions used in the experiment, and present the results in the revised Supplementary Fig. 2, as also described in our response to Comment 1 of the Reviewer. Comparing these results with the calculations for a continuous film, we see that all our conclusions are equally valid for 500 nm disks. To comply with the Reviewer's request, we also show in the Supplementary Fig. 2 the distribution of the static magnetization taking into account the demagnetization field. As can be seen from these data, the magnetization is nearly uniform throughout the disks and is well aligned with the external magnetic field, which is consistent with very small thickness-to-diameter ratio for these disks that determines the significance of the dipolar edge fields.

The Reviewer writes:

Experimental results taken for more than one specific field H are important as nonlinear damping depends critically on H .

Reply:

We performed measurements in a broad range of static fields, which showed qualitatively similar behaviors. We note that, in contrast to the resonant nonlinear damping, the non-resonant processes are not expected to critically depend on H . Complying with the Reviewer's comment, we have added a note on the field dependence on page 3 (note 24) of the revised manuscript. We have also added a reference to the recent paper on resonant nonlinear damping (Barsukov I. et al., Giant resonant nonlinear damping in nanoscale ferromagnets. arXiv:1803.10925).

Response to Reviewer #3

The Reviewer writes:

This is a very nice manuscript in which a solution to excite magnetization auto-oscillation from Spin Hall effect by suppressing non-linear damping is provided. The author have provided clear experimental evidence of complete damping compensation from spin current, and verified with micromagnetics simulation. The investigation clearly identify the cause to non-linear damping and the results will be very useful to develop functional spin nano-oscillators. The experiment and simulation are well designed and carried out. The manuscript is clear and is free of mistakes. I recommend the publication of this manuscript.

Reply:

We thank the Reviewer for the positive evaluation of our work and for the recommendation to publish our manuscript in its present form.

Reviewers' comments:

Reviewer #1 (Remarks to the Author):

I would like to thank for the detailed response of the authors to the previous comments. The updated manuscript has been improved with the conclusions being reinforced, and should be published. However, regarding to the additional measurements, I still have one concern.

Main comment:

In supplementary Fig. 3, the authors show the BLS peak intensity as a function of applied current for three CoNi samples with different amplitude of PMA. While the peak intensity of the sample with complete compensation between PMA and dipolar anisotropy is significantly stronger than the other two cases as $I > I_c$, the current corresponding to the maximum peak intensity is clearly postponed to a larger value for the CoNi samples in which the amplitude of PMA is larger or smaller than the dipolar anisotropy. This behavior is quite different from what has been observed in the case of Py. The authors should address this issue appropriately at least in the supplementary material, since the onset of the decrease of peak intensity is an important point, which implies the occurrence of a new relaxation process related to the nonlinear mode. Besides, I think it will also be useful, if the authors provide the data of current dependent spectral linewidth for all CoNi samples.

Minor comment:

In Line 163, Page 8, do the authors want to interpret that the frequency shifts for Py and CoNi samples are consistent between simulation and experimental results, indicating the dominant role of lowest-frequency mode? If so, it is better to reorganize these sentences, and make it clearer.

Reviewer #2 (Remarks to the Author):

The authors have significantly improved the presentation of their intriguing work. The added data and figure strengthen the discussion in a convincing manner. The authors have adequately responded to all comments and criticism. The manuscript warrants publication.

Response to Reviewer #1

The Reviewer writes:

I would like to thank for the detailed response of the authors to the previous comments. The updated manuscript has been improved with the conclusions being reinforced, and should be published. However, regarding to the additional measurements, I still have one concern.

Reply:

We thank the Reviewer for the positive evaluation of our response and the recommendation to publish our manuscript after an additional small revision. As described in detail below, we have complied with all Reviewer's requests. We hope that the Reviewer will find the additional revision satisfactory.

The Reviewer writes:

Main comment:

In supplementary Fig. 3, the authors show the BLS peak intensity as a function of applied current for three CoNi samples with different amplitude of PMA. While the peak intensity of the sample with complete compensation between PMA and dipolar anisotropy is significantly stronger than the other two cases as $I > I_c$, the current corresponding to the maximum peak intensity is clearly postponed to a larger value for the CoNi samples in which the amplitude of PMA is larger or smaller than the dipolar anisotropy. This behavior is quite different from what has been observed in the case of Py. The authors should address this issue appropriately at least in the supplementary material, since the onset of the decrease of peak intensity is an important point, which implies the occurrence of a new relaxation process related to the nonlinear mode. Besides, I think it will also be useful, if the authors provide the data of current dependent spectral linewidth for all CoNi samples.

Reply:

We would like to emphasize that the behaviors mentioned by the Reviewer are consistent with our interpretation. In fact, the current value, at which the maximum intensity is achieved, is determined by two factors: (i) the value of the intensity of magnetization oscillations necessary for the onset of strong nonlinear relaxation and (ii) the rate, at which the intensity increases with the increase of the current above the threshold value. The intensity (i) depends on the efficiency of the nonlinear coupling determined by the degree of ellipticity of precession. In agreement with this picture, the samples with larger ellipticity exhibit the onset of the strong nonlinear relaxation at smaller intensities. The rate (ii) is also expected to depend on the efficiency of nonlinear mechanisms. Since nonlinear scattering counteracts the energy flow due to the injection of the pure spin current, in samples with larger ellipticity, the rate (ii) should be smaller, in agreement with the data of Supplementary Fig. 3a. Finally, in samples with very large ellipticity, such as the Py sample (Fig. 2 in the main text), the strong nonlinear relaxation develops at very small intensities at currents close to the threshold current, resulting in the complete suppression of the auto-oscillations.

Complying with the Reviewer's request, we have added the Supplementary Note 2 discussing these issues. We have also added in the Supplementary Figure 3b the data for the current-dependent linewidth for all CoNi samples. We would like to note that, in BLS measurements, the measured linewidth in the auto-oscillation regime is mostly determined by that of the interferometer (see page 6 of the main text). Therefore, the data about the linewidth can only be used for a qualitative analysis.

The Reviewer writes:

Minor comment:

In Line 163, Page 8, do the authors want to interpret that the frequency shifts for Py and CoNi samples are consistent between simulation and experimental results, indicating the dominant role of lowest-frequency mode? If so, it is better to reorganize these sentences, and make it clearer.

Reply:

We thank the Reviewer for pointing out this issue. We agree that the formulation of the idea in the original manuscript was not clear enough. We have rewritten the paragraph on page 8 to make it clearer.

Response to Reviewer #2

The Reviewer writes:

The authors have significantly improved the presentation of their intriguing work. The added data and figure strengthen the discussion in a convincing manner. The authors have adequately responded to all comments and criticism. The manuscript warrants publication.

Reply:

We thank the Reviewer for the positive evaluation of our response and the recommendation to publish the revised manuscript in its present form.

REVIEWERS' COMMENTS:

Reviewer #1 (Remarks to the Author):

The authors have addressed all my concerns, and now I can recommend publication.

Response to Reviewer #1

The Reviewer writes:

The authors have addressed all my concerns, and now I can recommend publication.

Reply:

We thank the Reviewer for the positive evaluation of our response and the recommendation to publish the revised manuscript in its present form.